# Fabrication of Bio-Based Gelatin Sponge for Potential Use as A Functional Acellular Skin Substitute

**DOI:** 10.3390/polym12112678

**Published:** 2020-11-13

**Authors:** Mior Muhammad Amirul Arif, Mh Busra Fauzi, Abid Nordin, Yosuke Hiraoka, Yasuhiko Tabata, Mohd Heikal Mohd Yunus

**Affiliations:** 1Department of Physiology, UKM Medical Centre, Jalan Yaacob Latiff, Bandar Tun Razak, Cheras, Kuala Lumpur 56000, Malaysia; p94583@siswa.ukm.edu.my (M.M.A.A.); m.abid.nordin@gmail.com (A.N.); 2Centre for Tissue Engineering and Regenerative Medicine, Faculty of Medicine, Jalan Yaacob Latiff, Bandar Tun Razak, Cheras, Kuala Lumpur 56000, Malaysia; fauzibusra@ukm.edu.my; 3Biomaterial Group, R&D Center, Nitta Gelatin Inc. 2-22, Futamata, Yao City, Osaka 581-0024, Japan; yo-hiraoka@nitta-gelatin.co.jp; 4Department of Biomaterials, Institute for Frontier Medical Sciences, Kyoto University, 53 Kawara-cho Shogoin, Sakyo-ku Kyoto 606-8507, Japan; yasuhiko@frontier.kyoto-u.ac.jp

**Keywords:** gelatin sponge, genipin, carbodiimide, biocompatibility

## Abstract

Gelatin possesses biological properties that resemble native skin and can potentially be fabricated as a skin substitute for full-thickness wound treatment. The native property of gelatin, whereby it is easily melted and degraded at body temperature, could prevent its biofunctionality for various applications. This study aimed to fabricate and characterise buffalo gelatin (Infanca halal certified) crosslinked with chemical type crosslinker (genipin and genipin fortified with EDC) and physicaly crosslink using the dihydrothermal (DHT) method. A porous gelatin sponge (GS) was fabricated by a freeze-drying process followed by a complete crosslinking via chemical—natural and synthetic—or physical intervention using genipin (GNP), 1-ethyl-3-(3-dimethylaminopropyl) (EDC) and dihydrothermal (DHT) methods, respectively. The physicochemical, biomechanical, cellular biocompatibility and cell-biomaterial interaction of GS towards human epidermal keratinocytes (HEK) and dermal fibroblasts (HDF) were evaluated. Results showed that GS had a uniform porous structure with pore size ranging between 60 and 200 µm with high porosity (>78.6 ± 4.1%), high wettability (<72.2 ± 7.0°), high tensile strain (>13.65 ± 1.10%) and 14 h of degradation rate. An increase in the concentration and double-crosslinking approach demonstrated an increment in the crosslinking degree, enzymatic hydrolysis resistance, thermal stability, porosity, wettability and mechanical strength. The GS can be tuned differently from the control by approaching the GS via a different crosslinking strategy. However, a decreasing trend was observed in the pore size, water retention and water absorption ability. Crosslinking with DHT resulted in large pore sizes (85–300 µm) and low water retention (236.9 ± 18.7 g/m^2^·day) and a comparable swelling ratio with the control (89.6 ± 7.1%). Moreover no changes in the chemical content and amorphous phase identification were observed. The HEK and HDF revealed slight toxicity with double crosslinking. HEK and HDF attachment and proliferation remain similar to each crosslinking approach. Immunogenicity was observed to be higher in the double-crosslinking compared to the single-crosslinking intervention. The fabricated GS demonstrated a dynamic potential to be tailored according to wound types by manipulating the crosslinking intervention.

## 1. Introduction

Skin wound, either acute or chronic, is a major healthcare burden around the world. This is clear with the size of the global advanced wound care market with USD 10.43 billion in 2019 and this is projected to reach USD 15.59 billion in 2027 [1]. Currently, the increasing global Muslim populations have brought along a great demand in halal-based products such as food, supplements and medical devices. As a result, efforts toward developing the niche market of halal wound care products are also underway [2].

Specific to the management of full-thickness wounds, the loss of the dermis layer results in the inability of the keratinocytes from the native epidermis layer to re-epithelialise. Hence, any loss of full-thickness skin of more than 4 cm requires a skin graft in order for it to regenerate efficiently [3]. However, a skin graft commonly presents with complications such as secondary infection and pain, particularly at the donor site [4]. 

An alternative to a skin graft is the use of skin substitutes made from various biomaterials [5]. Instead of only covering the wound, these skin substitutes interact with the wound environment to facilitate healing [6]. These interactions are usually achieved via utilisation of bioactive polymers as raw materials, fabrication of a microstructure that mimics the extracellular matrix (ECM) and incorporation of soluble bioactive molecules into the bioscaffolds. In essence, skin substitutes can act as both a provisional scaffold for cell migration and an advocator for the wound to be healed [7]. 

The ECM of the skin is comprised mainly of collagen [8]. Accordingly, collagen is widely used in the skin substitute application such as Integra^®^, Apligraft, Excellagen^®^, CollaWound and many more [9]. The challenge in the production of collagen-based skin substitutes is the extraction of collagen from its natural source. In order to obtain intact collagen from specific animals’ tissue, extra precaution needs to be taken to prevent denaturation of the collagen conformation [10]. Consequently, the extraction of collagen can be complicated and may translate to a higher product cost. Down the line, the final product may be expensive and will be less accessible to the patient [11]. 

An alternative denatured collagen product known as gelatin can be effectively extracted from an animal by-product [12]. Gelatin shares many chemical and biological properties with undenatured collagen [13]. As a result, gelatin can serve as an alternative collagen with lower production costs. Moreover, manufacturing of halal gelatin can further improve the accessibility of the final product to the global Muslim population [2]. Furthermore, gelatin has been widely used in clinical settings including wound dressings, implantable antibiotic carriers, vascular stent modifying material and neurosurgical applications with desired biocompatibility, biodegradability and non-immunogenicity [14].

Nevertheless, a challenge ensues in utilizing both materials as skin substitute due to instability. This limits their applications in clinical settings [15]. As a result, crosslinking treatments are required to achieve the necessary stability via physical or chemical intervention. Physical crosslinking methods include dehydrothermal (DHT) treatment and exposure to ultraviolet radiation. The primary advantage of the physical methods is that they maintain the chemical composition of the scaffold and have rarely been reported to be toxic to the cell. However, the limitation of these methods is that they are less efficient for obtaining the desired amount of crosslinking [16]. 

Alternatively, many chemicals such as formaldehyde, glutaraldehyde, polyepoxy compounds, tannic acid, dimethyl suberimidate, carbodiimide and acyl azide have been used to chemically modify the gelatin for biomedical applications. Chemical crosslinkers provide more stable hydrolysis resistance and thermal stability; however, these synthetic crosslinking reagents are relatively toxic to the cell, hence indicating they are the least efficient in terms of biocompatibility [17]. The choice of crosslinking method in fabricating skin substitute requires a balance between the efficiency of the crosslinking and the potential cytotoxic effect to the resident cell in the wound site [18].

The key determining factors for developing a skin substitute that can facilitate the healing of a full-thickness wound is its physicochemical properties that are adaptive to the wound environment [19]. As a temporary scaffold, where the resident cells in the wound bed migrate and form newly formed skin at the injury site, it is important for the skin substitute to be degraded when the healing is complete [20]. Failure to degrade on time will result in further complications such as pain, infection and inflammation when the newly formed skin is aggregated in the scaffold [21]. Hence, biodegradability is a crucial physical property needed for skin substitute [19]. 

The freeze-drying method is widely used to fabricate 3D porous scaffolds, where the porosity of the scaffolds is achieved by the rapid freezing of polymer solution which results in the formation of ice crystals [22]. The microstructure of the scaffold influences the water absorption ability of the scaffold. The interconnected channels in a porous scaffold facilitate the movement of water [23]. On the other hand, a lesser degree of these interconnected channels creates resistance against water movement throughout the scaffold [24]. As a result, the scaffold has the ability to retain water. A wound that has high exudation requires high absorption ability in the skin substitute [25]. Alternatively, a low exudation wound requires the skin substitute to be able to retain water. Furthermore, the migration of local cells from the wound bed into the scaffold is also dependent on its porosity. 

Fortunately, the crosslinking treatment of biomaterial such as gelatin allows tunability. The biodegradability and microstructure of the gelatin scaffold (GS) can be adjusted according to its crosslinking degree [14]. The modification of the crosslinking degree can be achieved by selection of different types of crosslinker or adjustment of the crosslinker’s concentration. In turn, the crosslinking degree will determine the biodegradability rate of the GS [26]. Moreover, the crosslinking degree also determines the architecture of the scaffold, particularly its porosity. The scaffold porosity then determines the water absorbance or water retention ability of the scaffold. 

In recent years, increasing demand for a crosslinking agent that can form a stable and biocompatible crosslink product without causing cytotoxicity has brought attention to the natural crosslinking agent, known as genipin (GNP). GNP is a natural compound derived from gardenia fruit (Gardenia jasminoides ELLIS). GNP has gained increasing attention as a crosslinking agent following its excellent biocompatibility and ability to improve mechanical properties [27,28]. When gelatin is crosslinked with genipin, two mechanisms of reactions occur. The first reaction involves the amine group (lysine) of gelatin executing a nucleophilic charge ensuring heterocyclic linking of genipin to the gelatin amine. In the subsequent response, the ester group of the genipin undergoes nucleophilic exchange by the amine group of the second gelatin fragment, marking the beginning of the crosslinking [28]. From the occurrence of the reaction, long-range intermolecular crosslinks are made among the gelatin molecules (Figure 1A) [29].

Most of the known synthetic chemical crosslinkers stabilize a polymer by forming intermolecular bridges. This introduction of foreign molecules within the polymeric network typically results cytotoxicity in the crosslinked product [20]. Like GNP, the synthetic zero-length crosslinking agent, 1-ethyl-3-(3-dimethylaminopropyl) (EDC), can also form a stable and biocompatible product without causing cytotoxicity. This is due to the activation of EDC carboxylic acid residues conjugating with the adjacent amino groups instead of introducing foreign molecules into the network. As a result, EDC form intramolecular crosslinks within a gelatin molecule or short-range intermolecular crosslinks between two adjacent gelatin molecules (Figure 1B) [29].

Mechanical strength is a paramount consideration in the fabrication of a skin substitute. When the wound contracts, adequate mechanical strength is required to prevent the skin substitute from breaking and exposing the underlying tissue to external contamination [30]. A combination of GNP and EDC, the two crosslinkers with good biocompatibility and a distinct crosslinking mechanism, which may complement each other, is a crosslinking approach to fabricate GS with better mechanical strength (Figure 1C) [29]. Fortification of a genipin-crosslinked gelatin sponge with EDC may produce a skin substitute with better mechanical strength without jeopardizing its biocompatibility.

This study aims to develop a versatile gelatin-based sponge scaffold as a skin substitute for the treatment of full-thickness cutaneous wounds. The effect of GS fabricated via the freeze-drying method and crosslinked by GNP with or without fortification by EDC is investigated. Physicochemical, biomechanical and cellular biocompatibility and cell interaction with the fabricated GS, respective to each crosslinking method, were evaluated.

## 2. Materials and Methods 

This research was approved by Universiti Kebangsaan Malaysia (UKM) Research Ethics Committee (Code No. JEP-2018-0618) under the University Grant (Code No. GUP-2017-117).

### 2.1. Gelatin

Buffalo gelatin originally manufactured at a Nitta-Gelatin facility in Cochin, India was obtained from Nitta-Gelatin Ltd. (Osaka, Japan). A gelatin particle is negatively charged in physiological pH. The gelatin was extracted from buffalo raw bone materials and has halal certification from different regions. It came in a high-grade quality powder with a low endotoxin unit < 3000 that is essential to reduce immune rejection post implantation. 

### 2.2. Fabrication of GS

Gelatin powder was dissolved in distilled water for 30 min at 40 °C to achieve a final concentration of 5% (*w*/*v*). The gelatin solution was cast in a particular mould for further analysis. The solution was pre-frozen at −80 °C for 6 h before being transferred into the freeze-dryer for lyophilisation processing (Ilshin, Korea) for 48 h. 

Two types of chemical crosslinker were then prepared: Genipin (GNP; CBC, Touliu, Taiwan) and 1-ethyl-3-(3-dimethylaminopropyl) carbodiimide (EDC; Thermofisher, Rockford, IL, USA). The genipin solution was prepared by diluting genipin powder using 70% ethanol at two concentration: 0.1% (*w*/*v*) and 0.5% (*w*/*v*). The EDC solution was prepared by diluting EDC powder using distilled water and two concentrations were made: 15 mM and 30 mM. The fabricated GS was divided into five group. 

The first two were soaked into genipin solution at 0.1% (*w*/*v*) and 0.5% (*w*/*v*). The crosslinking process was stopped after 6 h, at which point the GS was removed and washed using phosphate-buffered saline (PBS). The GS was pre-frozen and lyophilized back for further analysis.

The other two-group crosslinking approach was a repetition of the previous procedure but it underwent a second crosslinked process using an EDC solution. GS with 0.1% (*w*/*v*) genipin was crosslinked again with 15 mM EDC, while GS with 0.5% (*w*/*v*) genipin was crosslinked with 30 mM EDC. All GS soaked with EDC solution underwent a 6 h crosslinking process before being removed and washed with 2-(N-morpholino)- ethanesulfonic acid (MES; pH 5.5). The GS was pre-frozen and lyophilized back for further analysis.

The last GS group was physically crosslinked using the dehydrothermal (DHT) method at 140 °C for 72 h. The crosslinked GS were then freeze dried again for 48 h for further analysis. A non-crosslinked GS was used as an experimental control.

### 2.3. Crosslinking Degree Evaluation

The crosslinking degree of fabricated GS was determined using a ninhydrin assay (Sigma-Aldrich, Saint Louis, MO, USA). The ninhydrin assay was used to determine the percentage of free amino groups in the gelatin samples, which can be calculated to the degree of crosslinking by comparing to uncrosslinked gelatin. The test samples were first lyophilized for 24 h and then weighed. Subsequently, the test sample was boiled with ninhydrin solution for 2 min at 100 °C following the manufacturer’s instructions. The amount of free amino groups in the test sample was determined using optical absorbance at 570 nm (Abs_570_) recorded with a spectrophotometer (BioTek, PowerWave XS, Highland Park, IL, USA). In addition, different concentrations (1.0, 2.0, 3.0, 4.0 and 5.0 mg/mL) of glycine were used as standards. The amount of free amino groups was proportional to the value of Abs_570_, whereas glycine at various known concentrations was used to create a standard curve of glycine concentration vs absorbance. The degree of crosslinking of the sample was then calculated following the equation below: (1)$$Degree\;of\;crosslinking=\frac{Amino_{0}-Amino_{c}}{Amino_{0}}*100$$
where *Amino*_0_ is the free NH_2_ concentration in non-cross-linked samples, and *Amino*_c_ is the free NH_2_ concentration in the cross-linked sample.

### 2.4. Enzymatic Biodegradation

The enzymatic hydrolysis resistance of GS was determined via incubation of each sample in 0.0006% (*w*/*v*) collagenase type I (Worthington, Lakewood, NJ, USA) at 37 °C for 14 h. A stop solution of 0.2 M ethylenediaminetetraacetic acid (EDTA) was added into the reaction followed by rapid cooling at 4 °C to terminate the enzymatic reaction. The GS was then subjected to the freeze-drying step followed by weighing of the dry weight. The dry weight of the GS was recorded every hour until the degradation was complete.

### 2.5. Thermal Stability

Thermal stability evaluation was measured by thermogravimetric analysis (TGA) that allows the measurement of the mass change of a sample similarly to the temperature changes in a controlled environment. The loss of weight of gelatin depends on its stability at different temperatures. Thermal stability of samples was measured with a thermogravimetric analyser. All tests were performed in a nitrogen-contained environment at a heating rate of 10 °C/min between 50 °C and 800 °C. The initial weight of each selected sample was approximately 10 mg.

### 2.6. Surface Topography and Microstructure Observation

The surface topography and cross-sectional microstructure of gelatin scaffolds were observed with scanning electron microscopy (SEM; FEI, Hillsboro, OR, USA). At days 1 and 7, the construct was fixed with 4% glutaraldehyde (Sigma, Saint Louis, MO, USA) and dehydrated through serial dilutions of ethanol and dried in a critical point dryer (Quorum, Q150R Plus, Laughton, UK). The samples were sputter-coated with gold and observed by scanning electron microscope (SEM) (SUPRA 55VP, Zeiss, Jena, Germany).

### 2.7. Pore Size and Porosity Evaluation

A microstructure analysis includes an evaluation of the pore size and porosity as described by Fauzi et al. (2019) [20]. The pore size of the scaffolds was arbitrarily quantified using Phenom Pro X integrated software (Phenom, Eindhoven, The Netherlands). The solvent replacement method was used to determine the porosity of the GS. Lyophilised GS were weighed (M_1_), immersed overnight in absolute ethanol (Merck, Darmstadt, Germany) and blotted with tissue paper to remove excess ethanol from the surface before weighing (*M*_2_). The dimension of the GS was determined using a Vernier caliper. Porosity was calculated using the following equation:(2)$$Porosity=\frac{\left(M_{2}-M_{1}\right)}{\rho V}$$
where *M*_1_ and *M*_2_ = masses of GS before and after immersion in absolute ethanol, respectively, *ρ* = density of absolute ethanol and *V* = volume of the GS. 

### 2.8. Swelling Ratio

The swelling ratio analysis was performed as described by Fauzi et al. (2019) [20]. Dry gelatin scaffolds were weighed (*W*_1_) and rehydrated in PBS for 2 h at room temperature. The excess buffer on the GS surface was blotted with tissue paper carefully and the wet weight (*W*_2_) was measured. The swelling ratios were calculated as follows:(3)$$Swelling\;Ratio\;\left(\%\right)=\frac{\left(W_{2}-W_{1}\right)}{W_{1}}*100$$

### 2.9. Contact Angle

Lyophilised gelatin scaffolds that were left at 20–24 °C overnight were used to determine the contact angle; 10 µL of distilled water was carefully dropped onto the surface of the gelatin scaffold, and images were captured using a digital camera with continuous shooting mode (Sony A6000, Tokyo, Japan). The contact angle was measured using Axio Vision LE image analysis software (Carl Zeiss, Dublin, CA, USA).

### 2.10. Water Vapour Transmission Rate

The water vapour transmission rate (*WVTR*) of the hydrogel was carried out as described by Mohamad et al. (2016) [31]. Prior to analysis, gelatin scaffolds were cut into disc-like shapes with 1.1 cm diameter and placed on the mouth of the glass vials containing 20 mL of distilled water. The vials were then placed in a humidity chamber at 37 °C with 84% humidity. Evaporation of water through the test membrane was monitored by weighing the vials at specific time intervals. Loss of water from the gelatin scaffold was determined by measuring the decrease in the weight. For analysis purposes, the data of reduced weight was plotted against time for each sample. Finally, the WVTR was determined by dividing the mean weight reduction of water by the area of the vial surface.
(4)$$WVTR=\frac{\Delta W}{At}$$

### 2.11. Energy Dispersive X-Ray

Energy dispersive X-ray spectrometry (EDS) analysis was performed using a Phenom Pro X SEM_EDX microscope (Phenom, Eindhoven, The Netherlands) to analyse the presence of elements in the scaffolds. Commercially available gelatin was used as a control. 

### 2.12. Fourier Transform Infrared Spectroscopy

The chemical characterisation of gelatin scaffolds was performed using Fourier transform infrared spectroscopy (FTIR). One mm^3^ of gelatin scaffolds was analysed, and the spectral data were recorded using a PE Spectrum 100 FTIR spectrometer (PE, Waltham, MA, USA) at a wavelength range of 700 to 4000 cm^−1^. The absorbance peaks were analysed to identify the chemical structure and changes resulting from the crosslinking.

### 2.13. X-Ray Diffraction Study

The X-ray diffraction (XRD) characterisation of the gelatin was performed using radiation at room temperature in the –2 scan mode. The diffraction pattern was recorded with an XRD analyser using CuKα radiation (λ = 1.542 Å) at 35 kV and 10 mA. The samples were scanned with 2θ (where θ is the Bragg angle) varying from 10° to 70° in a continuous mode. The result obtained was analysed using integrated software to identify the specific peaks.

### 2.14. Mechanical Properties Analysis

The mechanical strength of the gelatin scaffolds was measured in dry conditions at room temperature using Instron 8874 Tabletop Axial-Torsion Systems (Instron, Norwood, MA, USA) fitted with 50 N of load transducer at a crosshead velocity of 0.05 mm/min. Six scaffolds of 3 cm^2^ from each group were tested for evaluation using Young’s modulus and tensile strain. 

### 2.15. Cell Isolation and Culture

Redundant skin samples were obtained from consenting healthy patients undergoing abdominoplasty or circumcision. Sample collection was approved by Universiti Kebangsaan Malaysia (UKM) Research Ethics Committee (Code No. FF-2018-429). In brief, the skin sample (3 cm^2^) was cleaned of unwanted fragments such as fat, hair and debris, and minced into small pieces (approximately 2 mm^2^). Next, the skin was digested with 0.6% collagenase type I (Worthington, Lakewood, NJ, USA) for 5 to 6 h in a 37 °C incubator shaker followed by cell dissociation using 0.05% trypsin-EDTA (Gibco, Carlsbad, CA, USA) for 8–10 min. 

The digested skin containing both HEK and HDF was re-suspended in a co-culture medium at a 1:1 ratio (a mixture of HEK growth medium; Epilife^®^ (Gibco/BRL, Grand Island, NY, USA) and HDF growth medium; F-12:Dulbecco’s Modified Eagle Medium supplemented with 10% fetal bovine serum (FBS; Biowest, Riverside, MO, USA)) and seeded into three wells (surface area of 9.6 cm^2^/well) of a six-well culture plate (Greiner Bio-One, Monroe, NC, USA) at 37 °C with 5% CO_2_. The culture medium was replaced every two days until the cells reached the desired confluency. 

When the cells reached 70–80% confluency, differential trypsinisation was performed by using 0.05% Trypsin-EDTA for 3 to 5 min to dissociate the HDF from the culture surface. HEK and HDF were sub-cultured separately until the required number of cells was obtained, with the medium being changed every 48 h. Cells at passage 2 to 3 were used in all experiments.

### 2.16. Cell Seeding and In Vitro Analyses

Biocompatibility was evaluated via attachment, proliferation and cytotoxicity of HEK and HDF on GS. For cell attachment and growth rate analysis, HEK and HDF at passage 2 were seeded on the gelatin scaffolds at a density of 6 × 10^4^ cells/cm^3^ and 4 × 10^4^ cells/cm^3^, respectively. Unattached cells in the culture medium were quantified every hour for the first 6 h and 24 h after seeding using a trypan blue exclusion test. The percentage of cell attachment on gelatin scaffolds was measured using the following equation:(5)$$Cell\;attachment\;\left(\%\right)=\frac{\left(initial\;cell\;seeding-number\;of\;cells\;in\;DPBS\right)}{initial\;cell\;seeding}*100$$

The morphological features of the cells on gelatin scaffolds were observed using SEM. The proliferation of HEK and HDF on the gelatin scaffolds was analysed using a 3-(4, 5-dimethylthiazol-2-yl)-2,5-diphenyltetrazolium bromide (MTT) assay kit (Thermo Fisher Scientific, Waltham, MA, USA). Cells were cultured on GS, and the MTT assay was performed according to the manufacturer’s recommendations on days 1 and 7 to evaluate the growth rate. In brief, the culture medium was changed to 100 μL of fresh medium for each well; 10 μL of 12 mM MTT reagent (prepared with 1 mL of sterile PBS to one 5 mg vial of MTT) was added to the wells and incubated for 4 h at 37 °C. Then, 100 μL of dissolution reagents (10 mL of 0.01 M HCl to one tube containing 1 g of SDS) was added, followed by incubation for 4 h at 37 °C. The absorbance was measured using a spectrophotometer at 565 nm wavelength. 

The LIVE/DEAD^®^ Cell Viability Assay (Life Technologies, Carlsbad, CA, USA) was used to analyse the cytotoxic effect of gelatin scaffolds on HEK and HDF according to the manufacturer’s protocol. Briefly, HEK and HDF cultured on the gelatin scaffolds were incubated with 2 mM calcein AM and 4 mM EthD-1 in PBS for 30 min and washed with PBS prior to observation using Nikon A1R-A1 confocal laser scanning microscopy (CLSM; Nikon, Tokyo, Japan).

### 2.17. In Vitro Immunogenicity

Peripheral blood mononuclear cells (PBMCs) were obtained from healthy donors. The cells were separated by centrifugation in a Ficoll-Paque (GE, Marlborough, MA, USA) density gradient. PBMCs were stained with carboxyfluorescein succinimidyl ester (CFSE) as per the manufacturer’s instructions (BioLegend, San Diego, CA, USA). The cells were then resuspended in RPMI1640 (Invitrogen, Grand Island, NY, USA) containing 10% FBS. Subsequently, the cells were seeded at 2.5 × 10^5^ cells/cm^2^ and layered with 2 mm discs of gelatin scaffolds. The proliferative response of PBMCs was evaluated by assessing the incorporation of tritiated thymidine (3H-TdR; Perkin Elmer, Boston, MA, USA); 3H-TdR (0.037 MBq/well (0.5 μCi/well)) was added to the wells and incubated at 37 °C for 96 h. A total of 100 μL of cell suspension was transferred to a 96-well plate and exposed to a freeze/thaw cycle at −20 °C to lyse the cells, followed by harvesting onto a filter mat by using an automated cell harvester (Harvester Mach III M; TOMTEC, Hamden, CT, USA). Thymidine incorporation was measured in counts per minute via liquid scintillation spectroscopy on a beta counter (MicroBeta^®^ TriLux; Perkin Elmer, Waltham, MA, USA) after the addition of scintillation fluid (OptiPhase Super-Mix Cocktail; Perkin Elmer, Waltham, MA, USA). The cultures, activated with 1 μg/mL phytohemagglutinin (PHA), were used as a positive control. 

### 2.18. Statistical Analysis

Data were shown as mean ± SD. One-way analysis of variance (ANOVA) was used to compare the means of more than two groups. The comparison of the mean between the two groups was assessed with Student’s paired *t*-test. A *p*-value ≤ 0.05 was considered significantly different.

## 3. Results

### 3.1. The Stability of Fabricated Gelatin Bioscaffolds

The crosslinking of GS with a low concentration of GNP0.1 resulted in 39.2 ± 3.0% crosslinking degree. A high concentration of GNP0.5 increases the crosslinking degree to 50.7 ± 2.2%. A double crosslinking with GNP0.1EDC15 resulted in a significantly higher crosslinking degree compared to the GNP0.1 crosslinking alone at 57.5 ± 12.4% (*p* < 0.05). Similarly, double crosslinking with GNP0.5EDC30 resulted in a significantly higher crosslinking degree compared to the GNP0.5 crosslinking alone at 73.3 ± 4.3% (*p* < 0.05). Crosslinking with DHT demonstrated a comparable crosslinking degree with GNP0.1 at 46.7 ± 3.7% (Figure 2A).

Without crosslinking, the GS degrade completely in less than an hour when subjected to enzymatic hydrolysis. The crosslinking of GS with GNP0.1 resulted in a sponge that resisted enzymatic hydrolysis for 5 h. The increase of the GNP concentration to GNP0.5 resulted in increased resistance with which the GS resisted enzymatic hydrolysis for 10 h. Double crosslinking with GNP0.1EDC15 resulted in greater resistance compared to GNP0.1 crosslinking whereby the GS resisted enzymatic hydrolysis for up to 12 h. Double crosslinking with a high concentration of GNP0.5EDC30 demonstrated the greatest resistance whereby the GS lasted for up to 14 h. The lowest resistance was observed in the DHT crosslinking whereby the GS can only last up to 4 h (Figure 2B). 

In terms of thermal stability, the weights of the scaffolds changed with temperature (23–800 °C). Without crosslinking, the GS recorded a total weight loss of 60.02% at 800 °C. Crosslinking the GS with a low concentration of GNP (GNP0.1) reduce the total weight loss to 59.21%. An increase in the GNP concentration (GNP0.5) resulted in a decrease in the total weight loss to 57.34%. Double crosslinking with GNP0.1EDC15 resulted in a greater reduction of weight loss with 56.27% compared to GNP0.1 and GNP0.5 crosslinking. Double crosslinking at high concentration (GNP0.5EDC30) demonstrated the lowest total weight loss at 800 °C with 55.67%. Weight loss following DHT crosslinking was comparable to the GNP0.1 and the control (Figure 2C).

The mechanical strength of the gelatin scaffolds was evaluated based on Young’s modulus (stiffness) and tensile strain (elongation). Without crosslinking, the GS exhibited low stiffness (0.08 ± 0.01 GPa) and elongation (9.35 ± 1.86%). Crosslinking the GS with a low concentration of GNP0.1 resulted in 0.10 ± 0.04 GPa stiffness and 13.74 ± 1.28% elongation. An increase in the GNP concentration to GNP0.5 resulted in a significant increase in elongation (18.41 ± 0.68%; *p* < 0.05) compared to the control but not in the stiffness (0.10 ± 0.02 GPa). Double crosslinking with GNP0.1EDC15 resulted in a significant increase of stiffness (0.15 ± 0.02 GPa) and elongation (18.99 ± 2.10) compared with the control (*p* < 0.05) but not with GNP0.1 or GNP0.5. Similarly, double crosslinking with GNP0.5EDC30 resulted in a significant increase of stiffness (0.16 ± 0.01 GPa) and elongation (20.88 ± 5.47) compared with the control (*p* < 0.05) but not with GNP0.1, GNP0.5 or GNP0.1EDC15 crosslinking. DHT crosslinking demonstrated comparable stiffness and elongation to the GNP0.1 sponge (Table 1).

### 3.2. The Microstructure and Surface Characteristics of Fabricated Gelatin Bioscaffolds

Crosslinking of GS with GNP0.1 and DHT resulted in porosity of 78.6 ± 4.1% and 76.7 ± 6.3%, respectively. The differences were significantly less compared to the 96.1 ± 3.2% porosity in the control (*p* < 0.05). The porosity observed in GNP0.5, GNP0.1EDC15 and GNP0.5EDC30 treatment groups at 83.2 ± 4.4%, 86.0 ± 2.2% and 90.5 ± 4.1% was comparable to the control (Figure 3A). 

In terms of the pore size, crosslinking with GNP0.1 and DHT resulted in large pore sizes of 60–250 µm and 85–300 µm, respectively (Figure 3B). The pore sizes were relatively smaller in the control, GNP0.5, GNP0.1EDC15 and GNP0.5EDC30 treatment groups, with 40–150 µm, 60–150 µm, 50–150 µm and 40–150 µm, respectively.

The ability of a dressing to control water loss can be determined by the water vapour transmission rate (WVTR). Crosslinking of GS with GNP0.1 (247.6 ± 16.0 g/m^2^·day) and DHT (236.9 ± 18.7 g/m^2^·day) demonstrated a significantly lower WVTR compared to the control (282.0 ± 13.0 g/m^2^·day; *p* < 0.05). A high concentration of GNP0.5 resulted in a higher WVTR of 257.6 ± 5.8 g/m^2^·day. Double crosslinking with GNP0.1EDC15 resulted in an even higher WVTR at 267.9 ± 9.9 g/m^2^·day. Finally, double crosslinking at a high concentration of GNP0.5EDC30 resulted in a WVTR comparable to the control at 284.1 ± 5.9 g/m^2^·day (Figure 3C).

The contact angle is essential for measuring the ability of liquid–solid adhesion for bioscaffold fabrication. The contact angle provides the hydrophobic or hydrophilic characteristic of the sample’s surface. A lower degree contact angle (<90°) of the surface allows high wettability. All crosslinking methods resulted in a contact angle of less than 90° (Figure 3D). The sponge crosslinked with GNP0.1 and DHT demonstrated contact angles that were significantly higher than the rest of the treatments at 72.2 ± 7.0° and 73.6 ± 5.2°, respectively (*p* < 0.05). Double crosslinking with GNP0.1EDC15 exhibited a slightly lower contact angle (58.4.6 ± 1.8°) compared to GNP0.5 crosslinking (60.3 ± 2.8°). A high concentration of double crosslinking resulted in an even lower contact angle at 52.2 ± 1.5°.

Crosslinking of GS with a low concentration of GNP at GNP0.1 resulted in a 90.0 ± 6.3% swelling ratio compared to the control. A high concentration of GNP0.5 resulted in a lower swelling ratio at 84.8 ± 8.1% compared to the control. Double crosslinking with GNP0.1EDC15 resulted in an even lower swelling ratio at 82.1 ± 6.0% compared to the control. Only double crosslinking at a high concentration of GNP0.5EDC30 resulted in a significantly lower swelling ratio compared to the control at 78.8 ± 2.8% (*p* < 0.05). Crosslinking with DHT demonstrated a comparable swelling ratio with GNP0.1 at 89.6 ± 7.1% (Figure 3E).

### 3.3. Chemical Structure of Fabricated Gelatin Scaffolds

IR spectra of fabricated gelatin scaffolds demonstrated the FTIR spectra of the gelatin in the 900–1800 cm^−1^ finger-point region of wave numbers. For all of the scaffolds, the absorption peak at 3295 cm^−1^ was attributed to the stretching vibration of O-H, whereas the peaks at 1634 and 1383 cm^−1^ were attributed to the N-H stretching vibration. In the gelatin spectrum, the peaks at 1634, 1445 and 1375 cm^−1^ were attributed to amide I, amide II and amide III stretching vibrations. The result demonstrated that no change was observed at the amide II peak that confirmed the preservation of triple helix integrity after the crosslinking application (Figure 4A). 

X-ray diffraction (XRD) analysis elucidates the change in the crystal structure of the gelatin. The diffractogram of all GS appears as an amorphous material with no change towards the formation of the crystallinity after crosslink. The gelatin-based scaffolds have an intermediate characteristic reflection at 2θ = 20°. The results indicated that the control and all crosslinked scaffolds exhibit no major change of the gelatin crystal structure post-crosslinking (Figure 4B).

The elemental contents of gelatin scaffolds were evaluated using energy dispersive X-ray spectrometry (EDX) as shown in Table 2. There was no significant difference between each element among the scaffolds. It was noticeable that crosslinking of gelatin scaffold resulted in a slight decrease in carbon and a slight increase in oxygen elements; however, no significant difference was observed.

### 3.4. Cytotoxicity and Immunogenicity of Fabricated Gelatin Bioscaffolds

HEK (Figure 5A) and HDF (Figure 5B) culture remain viable when cultured on GS crosslinked with either GNP0.5 or GNP0.5EDC30. However, the double-crosslinked scaffold exhibited a slight cytotoxicity effect with the presence of dead cells in both HEK and HDF cultures, at day 1 (D1) or day 3 (D3). 

In regard to immunogenicity, culturing PBMCs on GS triggered the proliferation, in comparison with the negative control (Figure 5C). However, crosslinking GS, with GNP0.5 or GNP0.5EDC30, resulted in a lower proliferation rate compared to PHA activation. PBMC proliferation was higher when cultured with the double-crosslinked GS (41.4 ± 2.0%) compared to the single-crosslinked GS (22.7 ± 2.6%).

In terms of cell scaffold interaction, the GNP0.5EDC30 double crosslinking maintains a comparable cell attachment (Figure 6A,B) and growth rate (Figure 6C,D) compared to the GNP0.5 crosslinking alone. The representative SEM images of HEK (Figure 6E) and HDF (Figure 6F) cultured on GS revealed the infiltration of both cells in all sponges.

## 4. Discussion

This study demonstrated the fabrication of versatile sponge scaffold using gelatin for the treatment of full-thickness skin wounds. The Gelatin sponge (GS) biodegradability, thermal stability, mechanical strength, porosity, water retention ability, wettability and water absorption ability can be manipulated to suit the requirement of wound healing via the selection of crosslinking agents and the concentration. Fortification of genipin (GNP) crosslinking with 1-ethyl-3-(3-dimethylaminopropyl) carbodiimide (EDC) demonstrated the improvement of mechanical strength compared to crosslinking with GNP alone.

Both GNP and EDC crosslink the amine group within gelatin to either the other amine group or the carboxyl group. Thus, detection of the amount concerning the free amine group with a colorimetric indicator such as ninhydrin provides the information needed on how much of the polymeric network has been crosslinked in a bioscaffold [26]. This parameter, known as degree of crosslinking, is important as the higher the crosslink, the higher the ability of the scaffold to maintain its structure. As expected, an increasing trend for the degree of crosslinking was observed in this study following an increased number of crosslinking agents and their concentration. The degree of crosslinking can serve as a predictor towards all the favorable physical parameters required in the fabrication of the best skin substitute.

Crosslinking stabilizes the structure of a polymer and prevents it from degradation [17]. The stability of fabricated gelatin bioscaffolds is dependent to the crosslinking degree. In the current study, the highest concentration of double crosslink scaffolds unraveled higher enzymatic hydrolysis resistance and thermal stability compared to the remaining crosslinked group. This translates to a slower biodegradation rate when the skin substitute being implanted into the wound bed [32]. Lower degradation of double crosslinked scaffolds could be useful for cell growth and tissue regeneration [33]. In general, wound healing requires 3–4 weeks to be completed [34]. By manipulating the crosslinking process, the skin substitute can be fabricated to degrade according to the requirement of the specific individual skin wound [20]. 

It has been demonstrated that crosslinking of gelatin scaffold with GNP significantly improves mechanical strength in comparison to the non-crosslinked gelatin scaffold. This is due to the intermolecular connectivity that is created by the crosslinking process [35]. As expected, the addition of EDC crosslinking resulted in greater mechanical strength compared to GNP alone due to the higher intermolecular bridge formation produced by the two crosslinkers [20]. Sufficient mechanical strength is paramount for a skin substitute. This is to prevent the skin substitute from breaking and exposing the underlying tissue to external contamination when the wound contracts [30].

Fabrication of GS using the freeze-drying method was effective in producing a highly porous bioscaffold that is suitable to be used for formation of tissue substitutes. High porosity ensures cell penetration, adequate diffusion of nutrient and oxygen that leads to increased cell proliferation and vascularization during skin tissue regeneration [23]. However, without crosslinking, the aforementioned favourable microstructure is impractical as gelatin rapidly degrades at physiological temperature.

When the GS was crosslinked, the porosity observed in the control reduced. This is due to the crosslinking method that involved a washing step to remove residual crosslinker from the sponge. The porosity reduced significantly in low concentration of GNP and DHT method due to the incomplete crosslinking of the porous GS, resulting in non-crosslinked gelatin being cleared during the washing step [36]. However, from a recent systematic review that examines the different methods used in preparing genipin crosslinked-gelatin scaffold, the reduced porosity level is still within the reported 60% to 95% porosity range that displayed optimum functional parameters of GS for different tissue engineering application [37].

Freezing temperature during the freeze-drying process plays a crucial role in controlling the size of the ice crystal, and in turn pore size, porosity, interconnectivity and mechanical strength [23]. Freezing temperature of –80 °C has been reported to produce small pore size with high open porosity in gelatin scaffolds [28]. Consequently, the pore size observed was within the 100 µm to 300 µm range recommended to allow cell infiltration and capillary formation for tissue regeneration [38]. In a practical context, there is no standardized pore size nor does the ratio of opened and closed porosity exist as these requirements are heavily tissue dependent [38]. 

The addition of crosslinking resulted in an increase in pore size. Similar to porosity, incomplete crosslinking in the single GNP crosslink group and the physical crosslink group resulted in a large portion of non-crosslinked gelatin being washed away during the washing step. Nevertheless, the pore size in all of the crosslinked group is still within the aforementioned recommended range [38].

The higher the porosity, the greater the water absorption capacity of a bioscaffold. Alternatively, lower porosity represents higher water retention of a bioscaffold. In this study, the increase in scaffold porosity coincides with the increase in water absorption capacity as depicted by the swelling ratio. Alternatively, water vapor transmission and wettability, two parameters that represent water retention, were reduced with a lower scaffold porosity. 

Ultimately, the crosslinking process can be manipulated to fabricate a precise water absorption ability, water retention and wettability of the skin substitute depending on the requirement of the wound [39]. For example, high concentration double crosslinking can be utilized in a non-exudating wound where moisture retention plays a big role in the wound healing process. Alternatively, less crosslinking agent with lower concentration might be more beneficial in an exudating wound.

Cellular biocompatibility of scaffolds is a top precedence to ensure the success of gelatin scaffolds in the formation of skin substitute [19]. HEK and HDF, the two cells that reside in skin tissue, were used to evaluate the biocompatibility of GS to be used as a skin substitute. Cells seeded on the EDC crosslink scaffolds exhibited minimal cytotoxic effect as indicated by the live/dead assay. Additionally, the in vitro immunogenicity assay revealed a reduction in the cell cycle of peripheral blood mononuclear cells (PBMC) by EDC crosslinking.

The possible cytotoxicity effect of the EDC crosslinked gelatin might be attributed to the amount of urea byproduct [40]. The urea was shown to delay the cell cycle and cause cell death; however, the mechanism involved remains unclear [41]. Similarly, the cell cycle of PBMC was halted in EDC group treatment. Besides, it was assumed that reduction of open porosity and pore size may contribute to the retention and incomplete removal of EDC byproducts (urea) that might lead to cell death [20]. 

The GNP crosslink scaffold displayed better biocompatibility compared to the EDC as indicated by the absence of dead cells in the biocompatibity assay. However, cell attachment, proliferation and migration of HEK and HDF were comparable between GNP and EDC crosslinking scaffolds [20,42].

Limitations in this study include the exemption of the EDC-only crosslinking groups. This is due to the objective of utilizing EDC as a secondary crosslinker to reinforce GNP crosslinking without having adverse effects such as cytotoxicity. However, fortification of GNP with EDC demonstrated a hint of cytotoxicity compared to the GNP crosslinking. Fortunately, none of the trends reported in the biocompatibility parameters was statistically significant.

## 5. Conclusions

This study revealed the tunability of gelatin sponge scaffold as a potential skin substitute in the treatment of full-thickness wounds by manipulating the crosslinking agents and the concentration. The selection of crosslinking with genipin and 1-ethyl-3-(3-dimethylaminopropyl) carbodiimide (EDC), demonstrated good biodegradability, water retention ability, wettability and water absorption ability compared to genipin alone; as a result, the scaffolds obtained may be used for the manufacturing of different tissues in regenerative medicine regarding the suitability.

## Figures and Tables

**Figure 1 polymers-12-02678-f001:**
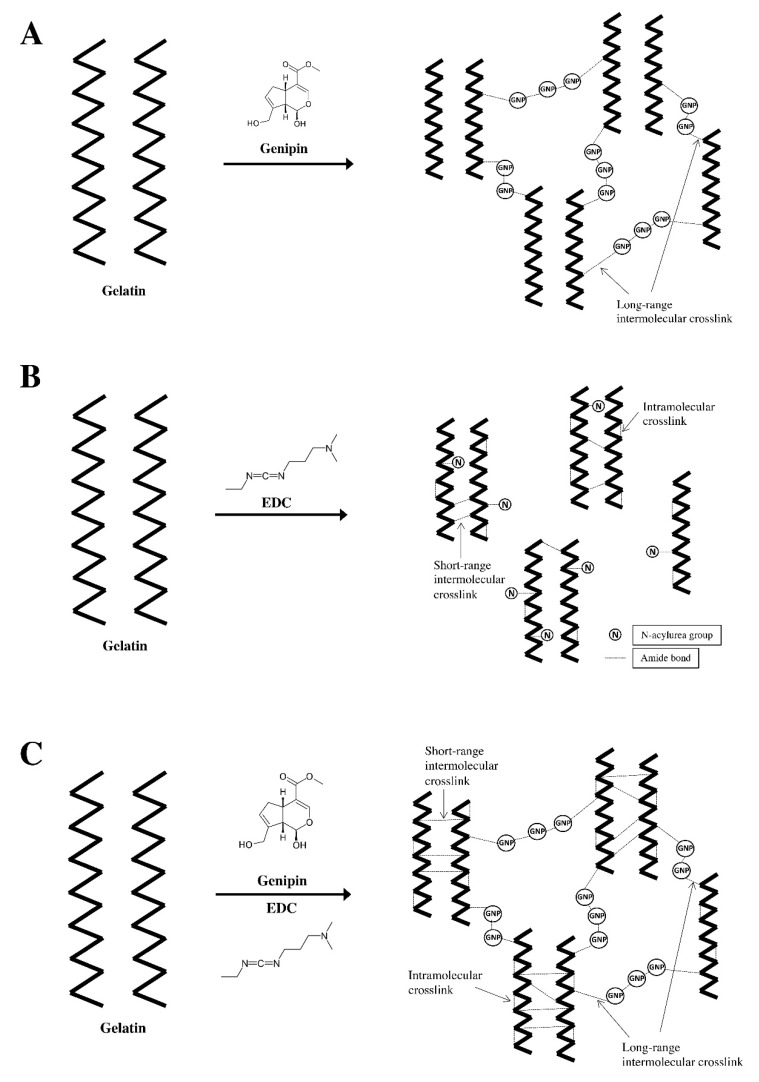
Crosslinking structure of (**A**) gelatin with genipin, (**B**) gelatin with EDC and (**C**) gelatin with double crosslinking of genipin and EDC.

**Figure 2 polymers-12-02678-f002:**
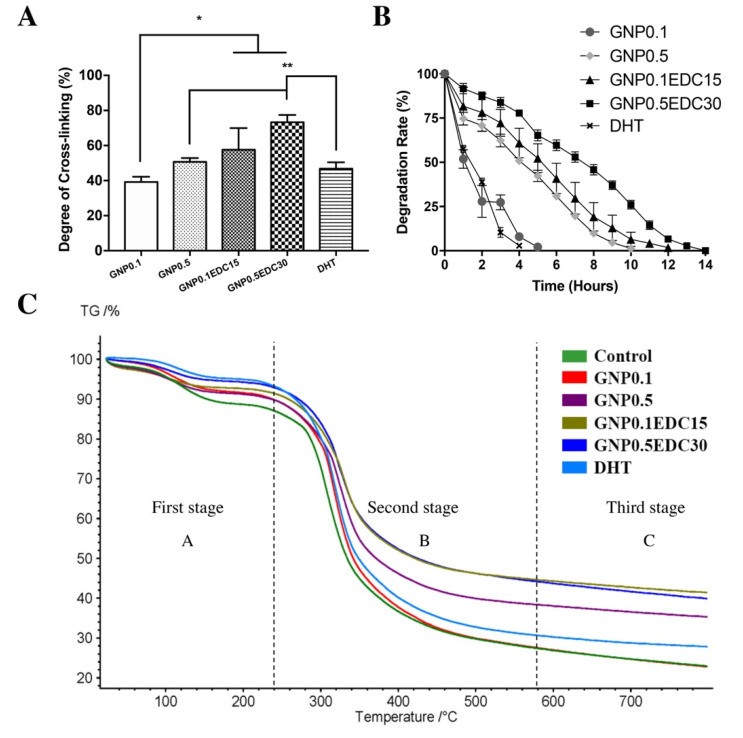
Effect of crosslinking on gelatin sponge (GS) biostability. (**A**) Measurement of crosslinking degree using ninhydrin assay. Double crosslinking and a higher concentration of crosslinker resulted in a higher crosslinking degree. Significant difference was observed between GNP0.1 and the DHT group with the other group; */** represent a significant difference (*p* < 0.05; n = 3, N = 5) between groups. (**B**) GS degradation under enzymatic hydrolysis. Double crosslinking and higher concentration of crosslinker resulted in longer degradation time. (**C**) Mass change of GS as a function of temperature. Double crosslinking and higher concentration of crosslinker resulted in reduced total.

**Figure 3 polymers-12-02678-f003:**
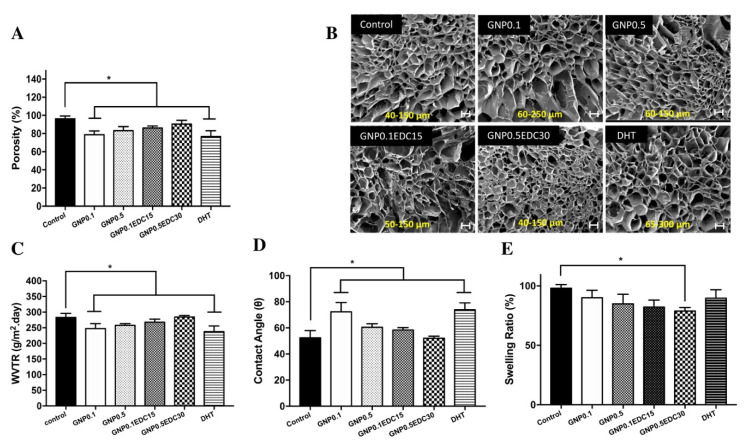
The effect of crosslinking on GS porosity. (**A**) Level of porosity of gelatin scaffold based on density displacement. Crosslinking resulted in reduction of porosity compared to control. (**B**) Representative SEM images of GS microarchitecture. Pore size range is depicted in yellow font. Double crosslinking and higher concentration resulted in identical microstructure with the control (30X magnification)/scale bar 100 µm. (**C**) WVTR measures water passage through the GS. Double crosslinking and higher concentration of crosslinker resulted in higher permeability. Only GNP0.1 and DHT demonstrated significantly lower permeability compared to the control as tested with Student’s *t*-test. (**D**) A lower contact angle indicates higher surface wettability of the GS. Double crosslinking and higher concentration of crosslinker resulted in higher wettability. Only GNP0.1 and DHT demonstrated significantly lower wettability compared to the control as tested with Student’s *t*-test. (**E**) Swelling ratio indicates the water absorption capability of GS. Double crosslinking and higher concentration of crosslinker resulted in lower water absorption. Only GNP0.5EDC30 demonstrated significantly lower water absorption compared to the control as tested with Student’s *t*-test. * represent a significant difference (*p* < 0.05; n = 3, N = 5) between groups.

**Figure 4 polymers-12-02678-f004:**
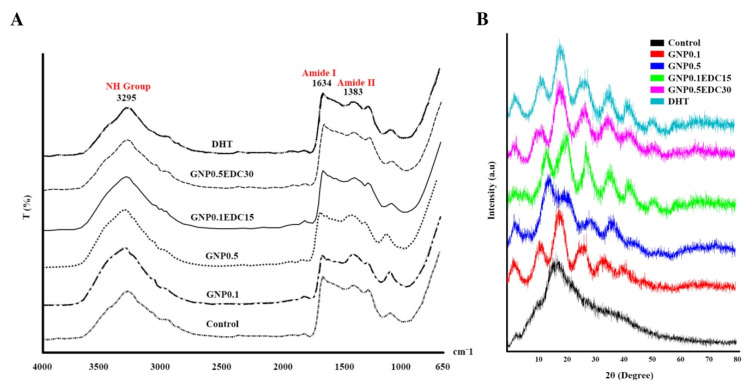
Effect of crosslinking on chemical composition and structure. (**A**) FTIR peaks indicate the chemical functional group detected in GS. Crosslinking did not alter the chemical functional group composition of gelatin. (**B**) XRD diffractogram indicates the crystal structure of GS. Crosslinking did not alter the crystal structure of gelatin.

**Figure 5 polymers-12-02678-f005:**
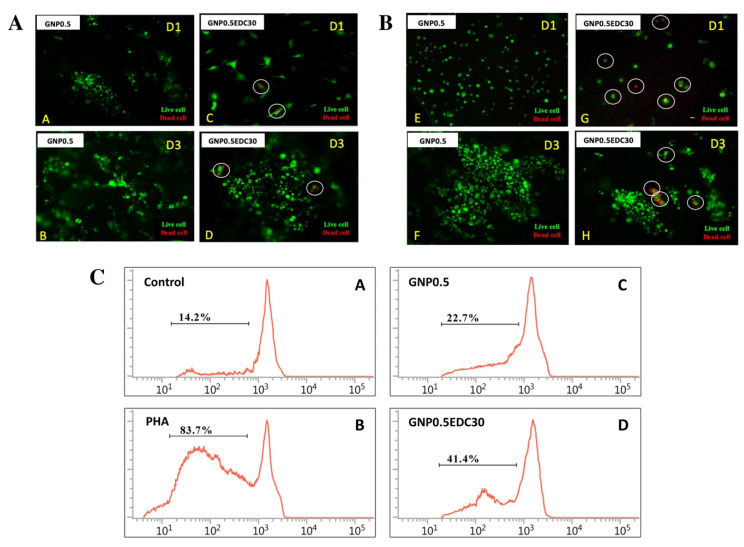
Effects of crosslinking on human epidermal keratinocytes (HEK) and dermal fibroblasts (HDF). Live/dead assay indicated the EDC crosslink scaffolds were toxic towards skin cells compared to genipin (GNP) and dihydrothermal (DHT) crosslink scaffold. (**A**) HDF cells and (**B**) HEK cells seeded on GS crosslink with GNP0.5/GNP0.5EDC30 at day 1 (D1) until day 3 (D3) (100X magnification). The white line circle indicates the dead cells. Effects of crosslinking on immunogenicity. (**C**) Proliferation properties of peripheral blood mononuclear cells (PBMCs) in medium supplemented with GS extracts in the carboxyfluorescein succinimidyl ester (CFSE)-labeling assay. (A–D) Representative FACS histograms of immune cells cultured with supplemented medium. Proliferation response without GS extracts was used as a control. PHA served as a positive control. PHA = phytohemagglutinin. The depicted line defines the level of proliferated immune cells. Results revealed that double crosslinking resulted in significant immune reaction in vitro. * *p* < 0.05 tested with Student’s *t*-test.

**Figure 6 polymers-12-02678-f006:**
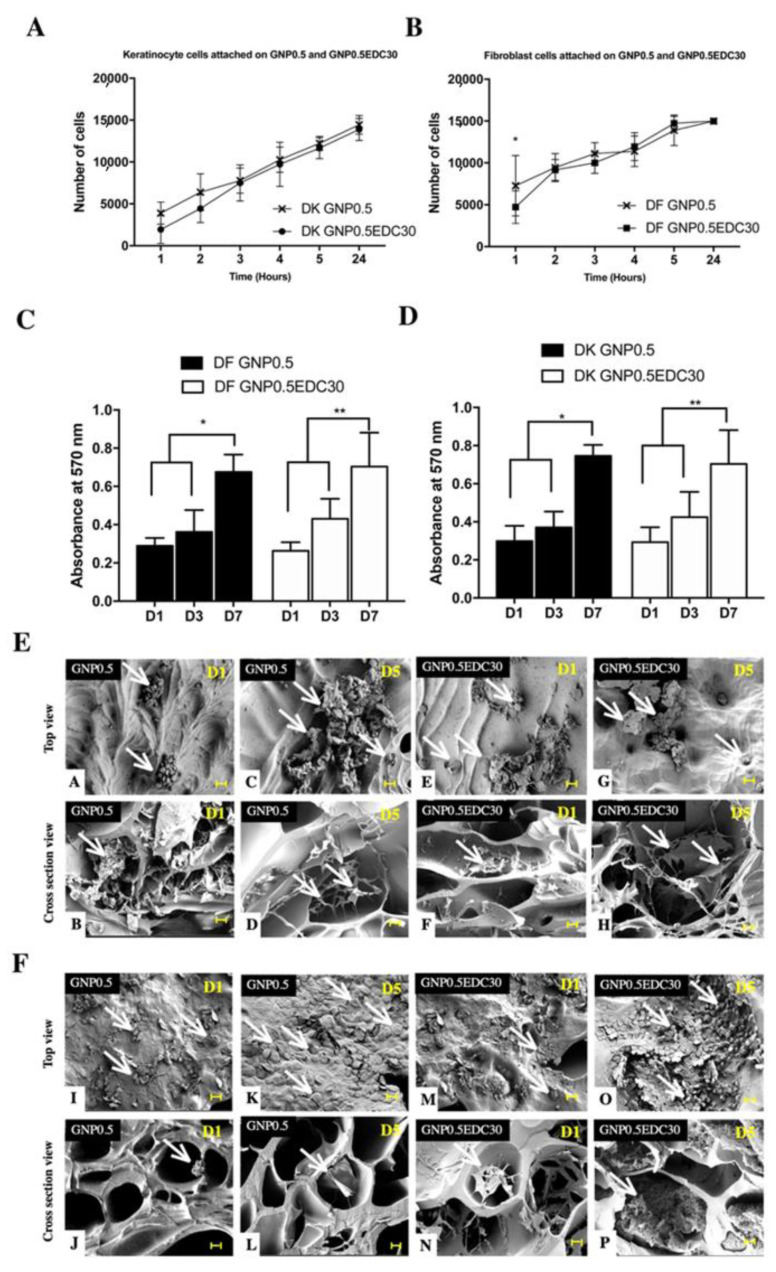
Cell attachment of (**A**) HEK and (**B**) HDF on GS crosslink with GNP0.5/GNP0.5EDC30 at 24 h. Significant difference at first hour of DF attachment are represented. Analysis of cell proliferation of (**C**) HEK and (**D**) HDF using MTT assay after 1, 3 and 7 days of cell seeding (n = 3). Cells seeded on both GNP0.5/GNP0.5EDC30 scaffolds proliferate efficiently after day 3. * *p* ≤ 0.05; ** *p* ≤ 0.01. Electron micrograph of crosslinked scaffold seeded with human dermal cell. (**E**) HDF cells and (**F**) HEK cells (I-P) seeded on GS crosslink with GNP0.5/GNP0.5EDC30 at day 1 (D1) until day 5 (D5) (250X magnification)/scale bar 20 µm are shown. White arrows indicate the cells attached.

**Table 1 polymers-12-02678-t001:** Mechanical strength of GS was evaluated through Young’s modulus and tensile strain.

Experimental Group	Young’s Modulus (GPa)	Tensile Strain (%)
Control	0.08 ± 0.01	9.36 ± 0.61
GNP0.1	0.10 ± 0.02	13.65 ± 1.10
GNP0.5	0.10 ± 0.01	18.35 ± 0.73 *
GNP0.1EDC15	0.15 ± 0.02 *	19.01 ± 1.31 *
GNP0.5EDC30	0.16 ± 0.01 *	21.26 ± 1.67 *
DHT	0.09 ± 0.01	13.88 ± 1.65

* significant difference (*p* < 0.05; n = 3) to control group.

**Table 2 polymers-12-02678-t002:** Elemental analysis by EDS. All scaffolds showed the presence of various elements such as carbon, oxygen and nitrogen which is the same as control.

Experimental Group	Elements (%)		
	C	N	O
Control	33.21 ± 1.75	36.44 ± 2.37	30.32 ± 1.09
GNP0.1	32.97 ± 1.89	29.19 ± 0.34	26.14 ± 5.76
GNP0.5	26.01 ± 6.45	28.97 ± 2.51	33.71 ± 3.48
GNP0.1EDC15	25.42 ± 0.65	39.37 ± 1.45	30.09 ± 1.40
GNP0.5EDC30	26.31 ± 1.16	30.03 ± 7.20	28.30 ± 1.39
DHT	25.42 ± 2.33	32.57 ± 4.78	31.23 ± 2.32

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
