# Peer review of "Fabrication of Bio-Based Gelatin Sponge for Potential Use as A Functional Acellular Skin Substitute"

_polymers, 2020, doi:10.3390/polym12112678_

Round 1

Reviewer 1 Report

This paper presents a series of GS scaffold on the basis of gelatin, which was crosslinked by genipin and EDC. Though the materials were new and the system was very detailed studied, there are still many problems need to be solved. I suggest this paper may be accepted and published on Polymers after the following questions can be answered.

  1. There is a huge contradiction in the system that according to the authors EDC is cytotoxicity, which disobey the key determining factors for a skin substitute, why the author choose EDC when there are many other crosslinking methods.
  2. Since EDC plays an important role in this system, why not mention it on the Abstract part?
  3. In the part of studying thermal stability, why choose 800 oC, at which the polymer chains are already carbonized?
  4. There are many repeated content on the parts of Introduction and Discussion, please make appropriate cuts. What’s more, the logic of the latter part is better than the former one.

Reviewer 2 Report

Journal: Polymers

Manuscript ID: polymers-970159

Title: Fabrication of Bio-Based Gelatin Sponge for Future Use as Functional Acellular Skin Substitute 

In this manuscript, the authors report the development of versatile gelatin-based sponge scaffolds as a skin substitute for the treatment of full-thickness cutaneous wounds. The authors, using a Buffalo gelatin, have fabricated freeze-dried scaffolds, in which gelatin was cross-linked either with neat genipin or genipin/1-Ethyl-3-(3-dimethylaminopropyl) carbodiimide (EDC) in different ratios/concentrations. They have also fabricated a cross-linked gelatin scaffold using a dihydrothermal method. Comparing the cross-linked scaffolds towards a pure gelatin scaffold they evaluate their physicochemical, biomechanical, cellular biocompatibility and cell-biomaterial interaction towards human epidermal keratinocytes (HEK) and dermal fibroblasts (HDF).

Recommendation: Major revision.

In general the manuscript needs a major revision in order to be suitable for publication.

Overall the language and grammar in the manuscript are not of a sufficient standard and need further work to improve it.

The title of the manuscript is confusing: “Fabrication of Bio-Based Gelatin Sponge for Future Use as Functional Acellular Skin Substitute”.

What the authors mean by “Future use”..??

Maybe the authors should discuss in the manuscript about the novelty of their study. Τhe genipin crosslinking of gelatin or EDC cross-linking are both widely studied in the literature. Moreover, if assume that the novelty lies on the combined genipin/EDC scaffold, the authors conclude that such a scaffold is not suitable as tissue engineered skin substitute. They report in their conclusions: “considering the negative effect of genipin and EDC crosslinking on biocompatibility compared to genipin alone, it is recommended to avoid using EDC as combine material for crosslinker in tissue engineered skin substitute”.

Why the authors did not include in their study gelatin scaffolds crosslinked only with EDC as well?

The discussion section needs a severe revision.  The authors give a lot of general information about the methods that they used or the importance of the experimental assays without really discussing the results of their work. It is the findings and their implications that they should widely discuss in this section.

There are many other issues as well that the authors should take into consideration to improve the manuscript.

For example:

-In Materials and Methods, the authors describe the fabrication of GS without mention in detail the experimental cross-linking conditions that they followed.

-In IR spectra as well, the authors report: “The result demonstrated that the FTIR spectra of composite scaffolds corresponded to gelatin crosslinked with GNP and EDC”.

However, the IR spectra seem quite similar to each other. How do the results demonstrate the crosslinking? The authors should explain. (The word “spectrophotometry” as well should be changed to "spectroscopy")

-In line 514, the authors report that “higher concentration of GNP as well as double crosslinking of GNP and EDC has been shown to increase the sponge porosity”.

Can the authors explain? Because it seems from the results that all the crosslinked sponges exhibited lower porosity compared to the control. (The results of the porosity measurements should be given into the Results Section as well.)

Figure 2. Authors should add a scale bar in SEM images.

Figure 4. it would be better to show calcein AM and EhtD images as well together with their merged images.

Round 2

Reviewer 2 Report

Journal: Polymers

Manuscript ID: polymers-970159 (2nd version)

Title: Fabrication of Bio-Based Gelatin Sponge for Future Use as Functional Acellular Skin Substitute 

In this new version, the manuscript is improved according to the suggestions of the reviewers but some issues are still remain that should be addressed by the authors.

Recommendation: Minor revision.

I still believe that the phrase “Future use” in the title of the manuscript is confusing: Maybe something like “potential use” could be more suitable, but I leave that to the authors to decide.

The language and grammar in the manuscript have been improved but they still need improvement:

Line 21: the word “crosslink” should be changed to “crosslinked”

Line 21: the word “has” should be changed to “had”

Line 29: “An increase in the concentration and double-crosslinking approach demonstrated an increment in the crosslinking degree, enzymatic hydrolysis resistance, thermal stability, porosity, wettability, and mechanical strength can be tunable differently from the control by approaching the GS via different crosslinking strategy.” The sentence is confusing..

Line 58: “were achieved” should be changed to “are usually achieved”

Line 70:  “shared” should be changed to “shares”

Line 86: “crosslinks” should be changed to “crosslinkers”

Similar changes should be made at:

Line 99: is achieved…that results….

Line 101: facilitate

Line 102: creates

Line 104: that has

Line 107: allows

Line 128: crosslikers

Line 129: result

Line 138: “GNP and EDC, the two  crosslinkers with a good biocompatibility and distinct crosslinking mechanism which may  complement each other” the sentence is confusing

Line 144: “The effect of GNP crosslinking with or without the fortification by EDC, following fabrication for GS via freeze-drying method is investigated.” The sentence is confusing.

Line 167: were soaked

Line 169: pre-frrezed

Line 171: underwent…crossliked

Line 172: crosslinked…soaked

Line 173: underwent….before removed..

Line 173: pre-freezed

Line 208: was measured

Line 372: ‘The increase of” the GNP

Line 437: demonstrated that..

Line 438:  “and these demonstrated” this phrase is confusing..maybe should be changed to “that confirmed”

Line 496: demonstrated the
